# PTPN1 Deficiency Modulates BMPR2 Signaling and Induces Endothelial Dysfunction in Pulmonary Arterial Hypertension

**DOI:** 10.3390/cells12020316

**Published:** 2023-01-14

**Authors:** Md Khadem Ali, Xuefei Tian, Lan Zhao, Katharina Schimmel, Christopher J. Rhodes, Martin R. Wilkins, Mark R. Nicolls, Edda F. Spiekerkoetter

**Affiliations:** 1Department of Medicine, Division of Pulmonary, Allergy and Critical Care Medicine, Stanford University, Stanford, CA 94305, USA; 2Vera Moulton Wall Center for Pulmonary Vascular Disease, Stanford University, Stanford, CA 94305, USA; 3Stanford Cardiovascular Institute, Stanford University School of Medicine, Stanford, CA 94305, USA; 4National Heart and Lung Institute, Hammersmith Campus, Imperial College London, London W12 0NN, UK

**Keywords:** PTPN1, BMPR2 signaling, hypoxia, endothelial dysfunction, pulmonary hypertension

## Abstract

Bone morphogenic protein receptor 2 (BMPR2) expression and signaling are impaired in pulmonary arterial hypertension (PAH). How BMPR2 signaling is decreased in PAH is poorly understood. Protein tyrosine phosphatases (PTPs) play important roles in vascular remodeling in PAH. To identify whether PTPs modify BMPR2 signaling, we used a siRNA-mediated high-throughput screening of 22,124 murine genes in mouse myoblastoma reporter cells using ID1 expression as readout for BMPR2 signaling. We further experimentally validated the top hit, PTPN1 (PTP1B), in healthy human pulmonary arterial endothelial cells (PAECs) either silenced by siRNA or exposed to hypoxia and confirmed its relevance to PAH by measuring PTPN1 levels in blood and PAECs collected from PAH patients. We identified PTPN1 as a novel regulator of BMPR2 signaling in PAECs, which is downregulated in the blood of PAH patients, and documented that downregulation of PTPN1 is linked to endothelial dysfunction in PAECs. These findings point to a potential involvement for PTPN1 in PAH and will aid in our understanding of the molecular mechanisms involved in the disease.

## 1. Introduction

Pulmonary Arterial Hypertension (PAH) is an obliterative disease of the pulmonary arteries that affects 50–100 individuals in one million people worldwide. The progressive increase in pulmonary vascular resistance ultimately leads to right heart failure, responsible for the high mortality in PAH. On a cellular level, pulmonary arterial endothelial cell (PAEC) dysfunction and apoptosis along with abnormal growth of smooth muscle cells (SMC) leads to medial thickening and neointima formation, causing the occlusive vasculopathy in PAH [1]. Understanding the molecular mechanisms that regulate the remodeling process of the vasculature is an area of intense study, yet no approved drug is available capable of reversing the remodeling, leaving PAH without a cure. Deficiencies of bone morphogenetic protein receptor 2 (BMPR2) expression and signaling are implicated in the development of PAH [2]. While BMPR2 mutations strongly predispose to PAH, only 20% of mutation carriers develop clinical disease, indicating that in addition to gene mutations, additional factors might be involved in the pathogenesis of PAH. Moreover, in many non-familial PAH forms, BMPR2 protein and signaling levels are reduced [2], suggesting that defective BMPR2 expression and signaling is a common phenomenon in different types of PAH. However, how the BMPR2 signaling is precisely regulated is largely unknown, especially in the non-genetic forms of PAH. Human clinical PAH features were observed in pulmonary endothelial-specific conditional BMPR2 knockout mice [3], SMC-specific BMPR2 dominant-negative mice [4,5], and BMPR2 heterozygous mutant rats [6]. Furthermore, defective BMPR2 signaling was shown to be linked with abnormal vascular cell phenotypes, such as abnormal proliferation, apoptosis, and angiogenesis of PAECs, and hyperproliferation and apoptosis resistance of pulmonary arterial smooth muscle cells (PASMCs) [2]. Increasing BMPR2 expression and signaling has therefore been proposed as an attractive solution with therapeutic potential for PAH treatment [2]. Indeed, our group has previously demonstrated that increasing BMPR2 signaling with Tacrolimus (FK506) [7] and BMPR2 expression with Enzastaurin [8] improved pulmonary vascular remodeling and PH in murine models of experimental PH. Most recently Sotatercept has been shown to effectively re-balance TGFb/BMPR2 signaling in preclinical models of PH as well as in a phase II PAH trial and was able to improve pulmonary vascular resistance in PAH patients on background PAH therapy (NEJM 2021), validating the approach of restoring normal BMPR2 signaling as a therapeutic approach in PAH.

Based on a high-throughput (HTS) siRNA screen of ~ genes, using a BRE reporter mouse cell line with ID1 expression as readout for increased BMPR2 signaling, in combination with an analysis of publicly available PAH RNA expression data, our group previously identified clinically relevant novel BMPR2 signaling modifier genes, namely, fragile histidine triad (FHIT) [8] and lymphocyte cell-specific protein tyrosine kinase (LCK) [9]. Using the same siRNA HTS data set as well as subsequent experimental validation in vitro, in vivo, and PAH clinical samples, we identified another novel BMPR2 signaling modifier gene, protein tyrosine phosphatase non-receptor type 1 (PTPN1), with potential relevance to PAH. In general, protein tyrosine phosphatases (PTPs) are involved in regulating signal transduction pathways and maintaining phospho-tyrosine levels in cells. Thereby they modulate a range of cellular processes, such as proliferation, differentiation, and apoptosis by working in concert with protein tyrosine kinases (PTKs) [10]. A recent study shows that the EYA3 tyrosine phosphatase activity promotes pulmonary vascular remodeling in PAH [11].

The PTPN1 encodes for PTP1b was the first-discovered PTP. In a case–control study, PTP1b single nucleotide polymorphisms (SNPs) were shown to contribute to hypertension [12]. Chronic insulin-mediated inhibition of PTPN1 function was found to upregulate platelet-derived growth factor (PDGF) signaling and to promote neointima formation in the balloon-injured rat artery [13]. PTPN1 knockout mice had increased mean arterial pressures [14]. Endothelial-specific PTP1b inhibition was shown to promote neointima formation in obese mice [15]. Adenoviral-mediated overexpression of dominant negative PTPN1 increased neointima formation in injured rat carotid arteries [16]. Furthermore, following vascular injury, mice deficient for PTPN1 in SMCs developed perivascular fibrosis in carotid arteries [17]. Using hypoxic PASMCs and hypoxia-induced PH mouse models, Freyhaus et al. demonstrated that hypoxia promoted PDGFRB pathway signaling through inhibiting PTPs, such as SHP-2, TC-PTP, DEP-1, and PTPN1 [18]. PTPN1 knockout mice showed exacerbated inflammation and leukocyte trafficking following ovalbumin challenge [19]. These findings suggested that PTP1b might play a significant role in pulmonary vascular remodeling and PAH. 

As endothelial dysfunction plays an important role in PH pathogenesis and as the BMPR2 receptor is highly expressed in endothelial cells, we aimed to investigate the role of PTPN1 in modulating BMPR2 signaling and its effect on PAEC function in PAH. We found that PTPN1 silencing with siRNA decreases BMPR2 signaling, which was associated with impaired proliferation, angiogenesis, and induced apoptosis in PAECs. We also found that PTPN1 RNA expression is downregulated in hypoxic PAECs, lung tissues of sugen5416/hypoxia PH rat models, and in whole blood samples of PAH patients by RNA sequencing. These findings point to a potential involvement for PTPN1 in endothelial dysfunction in PAH and will aid in our understanding of the molecular mechanisms involved in the disease.

## 2. Methods

### 2.1. High-Throughput siRNA Screen (HTS)

To identify novel BMPR2 signaling modifiers, a high-throughput siRNA screen of >22,124 genes was performed in an Id1-BRE luciferase containing a C2C12 mouse myoblastoma reporter cell line treated with or without BMP4, as previously described [8,9]. Following knockdown of the genes, viability of cells was assessed in cells stained with trypan blue staining. Changes in ID1 expression and cell viability after knockdown of the genes with siRNAs were calculated and compared with non-targeting siRNA-treated cells.

### 2.2. Cell Culture

Human healthy primary pulmonary arterial endothelial cells (PAECs) were purchased from PromoCell GmbH, Heidelberg, Germany (Cat # C12281). Cells were grown in standard endothelial growth medium (Cat # C-22120; PromoCell GmbH) with growth factors supplementation and 100 U/mL penicillin–streptomycin solution (Gibco) under standard conditions (37 °C, 5% CO_2_, 21% O_2_, 90% humidity). Cells were sub-cultured at 1:4 ratio and passages between 4 and 6 were used for all PAECs experiments. 

### 2.3. RNAi 

Knockdown of PTPN1 and BMPR2 was performed in PAECs. Briefly, PAECs (1.5 × 10^5^/well) were seeded onto 6-well plates and incubated at 37 °C in a humidified 5% CO_2_ atmosphere. Next day, cells were transfected with 50 nM siRNAs against PTPN1 (Cat # 4390824, Thermofisher, Waltham, MA, USA), BMPR2 (Cat # 4390824), and non-target (NT) controls (Cat # 4390843, Invitrogen), with 2 uL of Lipofectamine RNAimax (Cat # 13778-1.5, Invitrogen) in a total of 1 mL of OPTIMEM media. Six hours after transfection, the medium was changed to complete growth medium. After 48 h, knockdown efficiency was assessed by qRT-PCR. 

### 2.4. Cell Proliferation, Apoptosis, and Angiogenesis

Cell viability was assessed by MTT assay (Cat # V13154, Invitrogen, Waltham, MA, USA), as described previously [8]. Apoptosis was assessed by commercially available caspase 3/7 assay (Cat # G8090, Promega, Madison, WI, USA) as per the manufacturer’s instructions. Matrigel Tube formation assay was performed to assess angiogenesis, as described previously [9]. 

### 2.5. Hypoxia Induction

PAECs were grown under 1% O_2_ condition in a hypoxia chamber, as described previously [20]. Hypoxia induction was verified by measuring expression of hypoxia responsive gene, VEGFA mRNA expression by qRT-PCR. 

### 2.6. Sugen5416/Hypoxia-Induced PH Rat Models

PTPN1 mRNA and protein expression in the lung of sugen5416/hypoxia-induced PH experimental rat models was measured by qRT-PCR and western blotting, respectively. Pulmonary vascular remodeling and hemodynamic parameters of the rat samples chosen were previously reported in Dannewitz Prosseda et al. [8].

### 2.7. Gene Expression Quantification

Total RNA from PAECs was extracted and purified with commercially available RNeasy^®^ Plus Mini Kit (Cat # 74134, Qiagen, Hilden, Germany). Total RNA from rat lung tissues was isolated using Trizol extraction method, as described previously [21]. Total RNA was then converted to cDNA using a high-capacity cDNA reverse transcription kit (Cat # 4368813, Applied Biosystems^TM^, Foster City, CA, USA) and gene expression was assessed by TaqMan qRT-PCR with targeted TaqMan assay probes, human PTPN1 (Hs00942477), GAPDH (Hs01786624_g1), 18S (Hs99999901_s1), BMPR2 (Hs00176148_m1), ID1 (Hs03676575_s1), rat PTPN1 (Rn01423685_m1), rat Gapdh (Rn01775763_g1), and TaqMan™ 2× Universal PCR Master Mix (Cat # 4304437).

### 2.8. RNA-Seq Analysis of Whole Blood and PAECs from PAH Patients and Healthy Controls

To determine whether PTPN1 expression was downregulated in the blood and PAECs from PAH patients, we analyzed a large RNA-seq data of whole blood of patients with idiopathic, heritable, and drug-induced PAH (*n* = 359) compared to age- and sex-matched healthy controls (*n* = 72) [22] and a publicly available RNA-seq data set comprising 9 healthy and 9 PAH PAECs (GSE126262, [23]). We excluded data from one PAH patient due to the low quality of the RNA sequencing data compared to other PAH 8 samples. The subject characteristics include: PAH patients—2 HPAH (BMPR2 mutation), 5 IPAH, 1 APAH (drug and toxin associated), and 1 pulmonary veno-capillary disease, average age 35.78 years, 4 male/5 female, average 6MWD 301.43 m; and healthy controls—average age 43.22 years, sex 5 male/3 female.

### 2.9. Western Blotting

Western blotting was performed, as described previously [9], with primary antibodies PTP1b (Cat # CST5311, Cell Signaling, 1:1000), BMPR2 (Cat # MA5-15827, Invitrogen, 1:800), pSMAD1/5/9 (Cat # 13820S, Cell Signaling, 1:1000), ID1 (Cat # sc-133104, Santa Cruz, 1:100), and b-Actin (Cat # SC4778, Santa Cruz, 1:600), and secondary antibodies goat anti-rabbit IgG H&L HRP (Cat # ab6721, Abcam, 1:5000) and goat anti-mouse IgG H&L HRP (Cat # ab205719, Abcam, 1:5000).

### 2.10. Statistical Analysis

All data were analyzed using GraphPad prism version 9.0 (Graphpad Software, Boston, MA, USA). Data are represented as mean ± standard error mean. For data comparing between two groups, non-parametric student *t*-test was performed. *p* values < 0.05 were considered as significant changes. Number of samples for each experimental condition is presented in each figure and legend.

## 3. Results

### 3.1. PTPN1, a Novel BMPR2 Signaling Modifier

To identify novel modulators of BMPR2 signaling, we previously performed an unbiased siRNA-mediated HTS of 22,124 genes in a BRE-ID1-LUC reporter C2C12 mouse myoblastoma cell line. The screened hits were cross-validated in publicly available gene expression data sets of PBMC and lung tissues of PAH patients. These screening approaches identified three important BMPR2 modifying genes, FHIT [8], LCK [8,9] and Fyn [9], which play an important role in PAH. LCK and Fyn are PTKs. The phosphorylation of protein tyrosine is crucial to cellular signaling pathways. The level of protein tyrosine phosphorylation is regulated by PTKs and PTPs [24]. Several human diseases, such as cancers, are linked to aberrant tyrosine phosphorylation, and are associated with a dysbalance between PTK and PTP activity [24]. Here, we focused on the role of PTPs on BMPR2 signaling modulation. We assessed the select PTPs included in the HTS of 22,124 siRNAs for their potential to modulate ID1 expression (Figure 1A,B). There are over 100 PTPs in humans [25]; here, we focused on the receptor and non-receptor type PTPs, phosphatase of regenerating liver PTPs, map kinase phosphatases and atypical dual-specificity phosphatases PTPs, myotubularins PTPs, CDC14s, and class III PTPs in the HTS screening data set. Knockdown of the select PTPs showed inhibition of ID1 by several PTPs, such as PTPN1, PTPN11, and PTPRU, while several PTPs, such as PTPRS and PTPN20 showed increased expression of ID1 (Figure 1B). After re-testing of those PTPs that inhibited ID1 when knocked down (=ID1 stabilizing/activating PTPs), PTPN1 knockdown had the strongest effect on ID1 inhibition (Figure 1C). We then concentrated on PTPN1 on BMPR2 signaling modulation and validated these findings in PAECs. We found that PTPN1 knockdown with siRNA resulted in downregulation of BMPR2 and ID1 expression in PAECs after 48 h (Figure 1D–H). To determine whether PTPN1 was the upstream of BMPR2 signaling, we silenced BMPR2 with siRNA and found that PTPN1 expression was not altered (Figure 1I,J). This indicated that PTPN1 was upstream of the BMPR2 signaling. Together, these findings suggested that PTPN1 regulated BMPR2 expression and signaling in PAECs, as measured by ID1.

### 3.2. PTPN1 Inhibition Induces Endothelial Dysfunction in hPAECs In Vitro

Abnormal proliferation, apoptosis, and tube formation of PAECs is strongly linked with the pathogenesis of PAH [1,2]. We investigated whether PTPN1 regulates endothelial function in PAECs. We found that knockdown of PTPN1 with siRNA decreased cell viability, as measured by MTT assay and hemocytometer cell counts (Figure 2A). Compared to non-target siRNA controls, PTPN1 siRNA-mediated knockdown induced apoptosis, as confirmed by increased level of caspase3/7 activity (Figure 2B). We also observed that siRNA-mediated knockdown of PTPN1 reduced tube formation 6 h after cells were seeded in a Matrigel tube formation assay compared with cells treated with control non-target siRNA (Figure 2C). These data suggest that PTPN1 deficiency may lead to endothelial dysfunction in PAH.

### 3.3. PTPN1 Is Downregulated in Hypoxic hPAECs and in the Lung of the Sugen5416/Hypoxia/Normoxia Rat Model

Hypoxia is considered one of the contributing factors to Group 3 PH (PH linked with hypoxia and lung disease) and is used as an insult to produce experimental PH in rodents. Furthermore, a previous study described the regulation of PTP by hypoxia [18]. We therefore asked whether hypoxia exposure altered expression of PTPN1 in PAECs. PAECs were exposed to hypoxia for 72 h and PTPN1 expression was measured by qRT-PCR. Consistent with the previous findings in PASMCs [18], we found that hypoxia downregulated PTPN1 expression in PAECs (Figure 3A). Furthermore, we also measured PTPN1 mRNA and protein expression in the lungs of a sugen5416/hypoxia/normoxia-induced PH rat model by qRT-PCR and western blot, respectively. PH was induced in the rats that received a single subcutaneous dose of sugen5416 (a VEGF receptor tyrosine kinase inhibitor), followed by exposure to chronic hypoxia (10% O_2_) for 3 weeks and normoxia for 5 weeks (21% O_2_). The PH model was confirmed by measuring RVSP, RV hypertrophy (Fulton Index: weight ratio of the RV/left ventricle (LV) and septum (LV + S) (Please see the data in [8]). We found a trend towards lower PTPN1 mRNA and protein expression in the lung of sugen5416/hypoxia (SuHx) treated rats compared to normoxia-treated rats (Figure 3B,C), with a high variability between samples. Together, these findings suggest that PTPN1 deficiency may be associated with experimental, hypoxia-associated PH.

### 3.4. PTPN1 Is Downregulated in the Blood of PAH Patients and the Expression of PTPN1 Is Correlated with BMPR2 Signaling in PAECs of Healthy and PAH Patients

To determine whether PTPN1 expression was reduced in clinical PAH, first we measured PTPN1 expression in the whole blood of patients with idiopathic, heritable, and drug-induced PAH (*n* = 359) compared to age- and sex-matched healthy controls (*n* = 72) by RNA-seq. We observed a significantly downregulated PTPN1 expression in PAH patients compared to healthy controls. (Figure 4A), for subject characteristics, please see in [22]). Next, to determine whether PTPN1 expression was decreased in PAECs from PAH patients, we re-analyzed a publicly available RNA-seq data set comprising nine healthy and nine PAH PAECs (GSE126262, [23]). Although we did not observe changes in PTPN1 expression between PAH and healthy control PAECs, we found a significant correlation of PTPN1 expression with the BMPR2 signaling factors, BMPR2, SMAD5, and SMAD9 expression (Figure 4C–E). Correlation of PTPN1 with SMAD1 and ID1 was not statistically significant (Figure 5. These findings suggest the link of PTPN1 with BMPR2 signaling in PAH.

## 4. Discussion

Previously, several PTPs have been shown to be linked to the pathogenesis of PAH. For instance, pharmacological inhibition of Shp2, also known as PTPN11, ameliorated monocrotaline-induced PH-related hemodynamic parameters (mPAP, RVSP, and RVH) and improved pulmonary vascular remodeling in rats [26]. Xu et al. showed that inhibition of PTPRD in human PASMCs and rats resulted in PH through promoting PASMCs migration via the PDGFRB/PLCγ1 axis [27]. DUSP5-mediated inhibition of PASMCs proliferation suppressed PH and RVH [28]. Here, through a combined approach of HTS, in vitro validation and analysis of a large cohort of PAH clinical samples, we identified another PTP that is associated with PAH. We found PTPN1 as a novel modifier of BMPR2 signaling and showed that PTPN1 is decreased in the blood of PAH patients and PTPN1 deficiency is associated with induction of markers of endothelial dysfunction.

Loss of function mutation in BMPR2 occurs in 60–80% of familial cases of PAH patients, but the disease penetrance rate is low [29,30], suggesting that, in addition to the gene mutations, other unidentified genetic, epigenetic, or environmental factors may contribute to the pathogenesis of PAH. Importantly, defective BMPR2 signaling is also a common phenomenon in PAH patients regardless of their etiologies of the disease. However, the molecular mechanism of how BMPR2 signaling is precisely regulated remains unclear. Through a siRNA-mediated HTS and further experimental validation, we identified PTPN1 as a novel regulator of BMPR2 signaling. However, it remains unknown how PTPN1 regulates BMPR2 signaling. While we did not find relevant studies linking PTPN1 to BMP-SMAD signaling, a previous study performed in hepatocytes from PTPN1 knockout mice showed that those cells were resistant to TGFβ-induced downregulation of cell viability and upregulation of apoptosis [31]. The authors also revealed that PTPN1-deficient hepatocytes were less responsive to TGFb as a significant decrease of SMAD2/3 phosphorylation and increased NF-κB pathway activation in PTPN1 knockout hepatocytes was observed upon TGFß treatment [31]. Matulka et al. showed that PTPN1 was involved in the Activin/ALK4 signaling to modulate p-ERK1/2 signaling, which represents a noncanonical Activin pathway in embryonic stem cells [32]. While these studies support the role of PTPN1 in TGFß signaling, the precise mechanism by which PTPN1 might facilitate the TGFb/BMPR2 disbalance observed in PAH needs to be explored in future studies.

We found a significant downregulation of PTPN1 in the whole blood of a large cohort of PAH patients by RNA-seq analysis, yet the precise cell type or tissue responsible for the observed PTPN1 downregulation is not known. Previous reports describe a link between PTPN1 SNPs and systemic hypertension [12,33]. Hypoxia downregulates select PTPs including PTPN1 in PASMCs exposed to hypoxia and in the lung of hypoxia-induced PH mice [18]. Given that BMPR2 is highly expressed in endothelial cells, we herein investigated the role of PTPN1 in PAECs. In line with a previous report [18], we demonstrated a decreased expression of PTPN1 in PAECs exposed to hypoxia. While we observed a trend towards lower PTPN1 levels in the lung of sugen5416/hypoxia/normoxia treated rats with PH, the results were not significant, most likely due to the high variability of PTPN1 expressions in the lung samples. While hypoxia is used as an injurious stimulus in experimental PH, hypoxemia is usually not observed in PAH patients, in contrast to Group 3 PH patients. While the cause of PTPN1 downregulation remains to be determined, its effect on BMPR2 signaling and endothelial health could be an important contributor to PAH. We find that PTPN1 deficiency induces endothelial dysfunction by reducing proliferation, causing apoptosis and reducing tube formation of PAECs, all features linked to the development of PAH. Berdnikovs et al. furthermore demonstrated that PTP1B was involved in inflammation as inducible endothelial cell-specific deletion of PTP1B showed a significant increase in accumulation of eosinophils bound to the luminal surface of the endothelium in the lung vasculature and had a decrease in leukocyte recruitment into the lung tissue during ovalbumin-induced allergic lung inflammation [34]. Furthermore, in response to arterial injury, vascular smooth muscle cells deficient for PTPN1 promoted perivascular fibrosis [17]. During respiratory syncytial viral-induced exacerbation of chronic obstructive pulmonary disease, PTPN1 deficiency was shown to promote disease symptoms partly through enhancing S100A9 levels, a damage-associated molecular pattern molecule [35]. In contrast to these findings, several other lung studies show that PTP1b deficiency protected against lung inflammation. For instance, using polysaccharides (LPS)-induced acute lung injury (ARDS) models, Song et al. showed PTP1b inhibition protected against lung injury, potentially regulating through the CXCR4/PI3Kγ/AKT/mTOR signaling axis [36]. Whether PTPN1 deficiency protects or exacerbates lung remodeling is likely dependent on the context and animal disease model. As endothelial dysfunction and perivascular inflammation are key players in PAH pathobiology [37], maintaining adequate PTPN1 levels could be significant for pulmonary vascular remodeling in PAH.

While we observed a significant downregulation of PTPN1 expression in the whole blood of PAH patients, we did not find these changes in PAECs of PAH patients. This could be due to the fact of relatively lower expression of PTPN1 and variation in PAECs than in the blood cells. Further larger studies would require confirming these findings in PAECs of PAH patients. However, we found a significant correlation between PTPN1 and BMPR2 signaling marker gene expression in the data set (Figure 4C–E), indicating again the involvement of PTPN1 in BMPR2 signaling in PAH.

This study has several limitations. First, we demonstrated that inhibition of PTPN1 decreases BMPR2 expression and signaling and induces endothelial dysfunction; however, it would be important to see whether the opposite is true as well: whether overexpression of PTPN1 activates BMPR2 signaling and reverses abnormal endothelial cell behaviors. Second, this study lacks the direct causal link between PTPN1 and PAH in vivo. Further studies need to characterize whether global or cell-specific deletion of PTPN1 causes experimental PH or alternatively predisposes to a more severe PH phenotype in response to injurious agents used to induce experimental PH, such as hypoxia alone, sugen5416/hypoxia, or monocrotaline. Third, we were not able to validate in this pilot study the PTPN1 PAH blood RNA-seq expression data in a second sample cohort. Fourth, mechanisms of how PTPN1 regulates BMPR2 signaling in PAH remains to be explored. The crosstalk between BMPR2 signaling and the other signaling pathways, such as VEGF, ERK/MAPK, PI3K, AKT, p38, JNK, NOTCH, and TAK1, in the context of PTPN1 deficient cells would also need to be explored.

In summary, we find PTPN1 expression is downregulated in the whole blood of PAH patients and PAECs exposed to hypoxia and that PTPN1 downregulation is associated with endothelial dysfunction in PAECs. Furthermore, we discovered that PTPN1 is a novel modulator of the BMPR2 signaling pathway. These findings will help investigate the precise role of PTPN1 in PAH.

## Figures and Tables

**Figure 1 cells-12-00316-f001:**
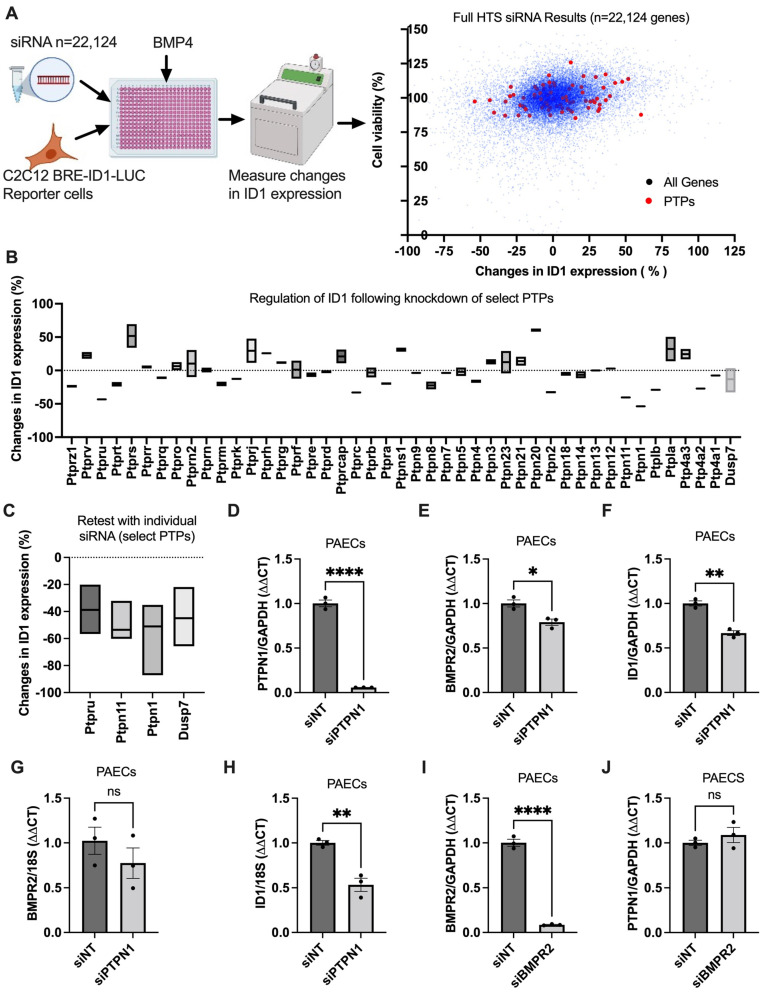
**PTPN1 is a novel regulator of BMPR2 signaling pathway.** A siRNA-mediated HTS (*n* = 22,124) was performed to identify possible BMPR2 signaling modifiers in mouse myoblastoma BRE-ID1-LUC incorporated reporter cells. After 48 h knockdown of the genes, cells were treated with BMP4 to activate the signaling for two hours and then measured for ID1-linked luciferase levels. A colorimetric trypan blue cell viability was also performed. (**A**) Changes in ID1-linked luciferin expression (*n* = 22,124) (*X*-axis) versus cell viability (*Y*-axis) were plotted. Red dots denote the pre-selected protein tyrosine kinases (PTPs) selected from all major PTPs. (**B**) Changes in ID1-linked luciferin expression of the selected PTPs (% changes from NTi). (**C**) Selected PTPs tested with individual siRNA (reconstructed from the secondary screening data *n* = 96 from [8]). Data represented as mean ± SEM (*n* = 2–3). (**D**–**J**) PTPN1, BMPR2 and ID1 expression were measured by qRT-PCR in PAECs silenced to either PTPN1 or BMPR2 for 48 h. Data represented as mean ± SEM (*n* = 3), student t-test. * *p* < 0.05, ** *p* < 0.01, **** *p* < 0.0001.

**Figure 2 cells-12-00316-f002:**
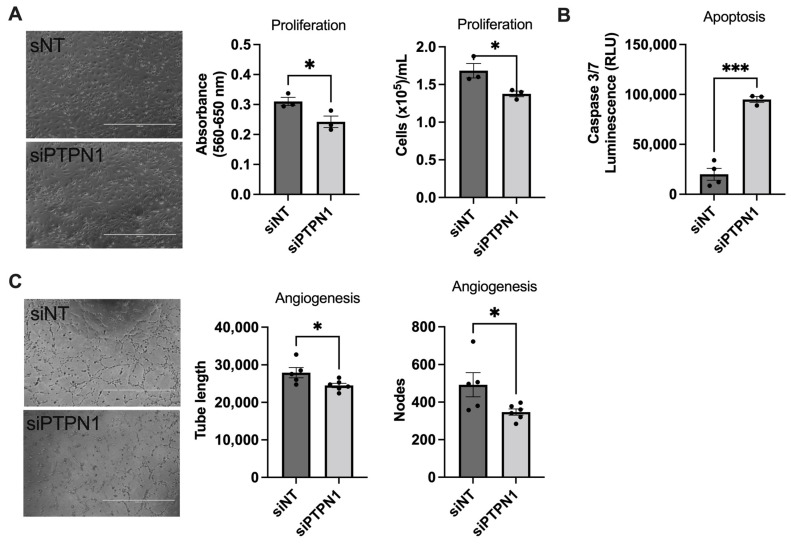
**PTPN1 deficiency induced endothelial dysfunction in PAECs.** (**A**) siRNA-mediated knockdown of PTPN1 showed decreased PAEC viability, as assessed by MTT assay and hemocytometer countings. (**B**) PTPN1 silencing induced apoptosis, as evidenced by increased caspase 3/7 levels in PAECs. (**C**) PTPN1 silencing decreased ability of PAEC tube formation in a Matrigel tube formation assay. Angiogenesis was quantified in the images using ImageJ software. Total tube lengths were presented in µm. The number of nodes were counted in the analyzed area. Scale bar = 1000 µm. Data represented as mean ± SEM (*n* = 3–6), student t-test. * *p* < 0.05, *** *p* < 0.001.

**Figure 3 cells-12-00316-f003:**
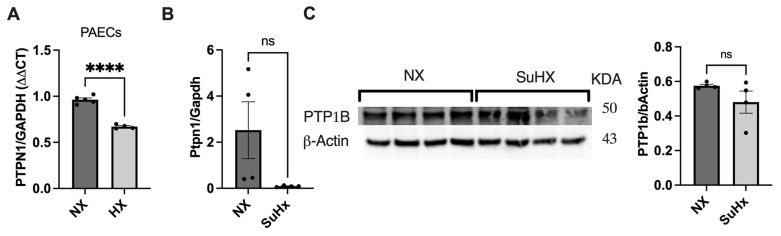
**PTPN1 is downregulated in healthy human PAECs exposed to hypoxia and in the lung sugen5416/hypoxia-induced PH rats.** (**A**) 150,000 PAECs were seeded onto 6-well plates and exposed to 72 h of hypoxia. After that, PTPN1 expression was measured by qRT-PCR. Induction of hypoxia was verified by measuring VEGF expression by qRT-PCR [20]. (**B**,**C**) PTPN1 mRNA and protein expression was also measured in the lung of sugen5416/hypoxia rate models by qRT-PCR and western blotting (please see the PH model description and phenotypes data in [8]). Data are represented as mean ± standard error mean, *n* = 3–5; student *t*-test was performed to compare data between two samples. **** *p* < 0.0001. ns, not significant.

**Figure 4 cells-12-00316-f004:**
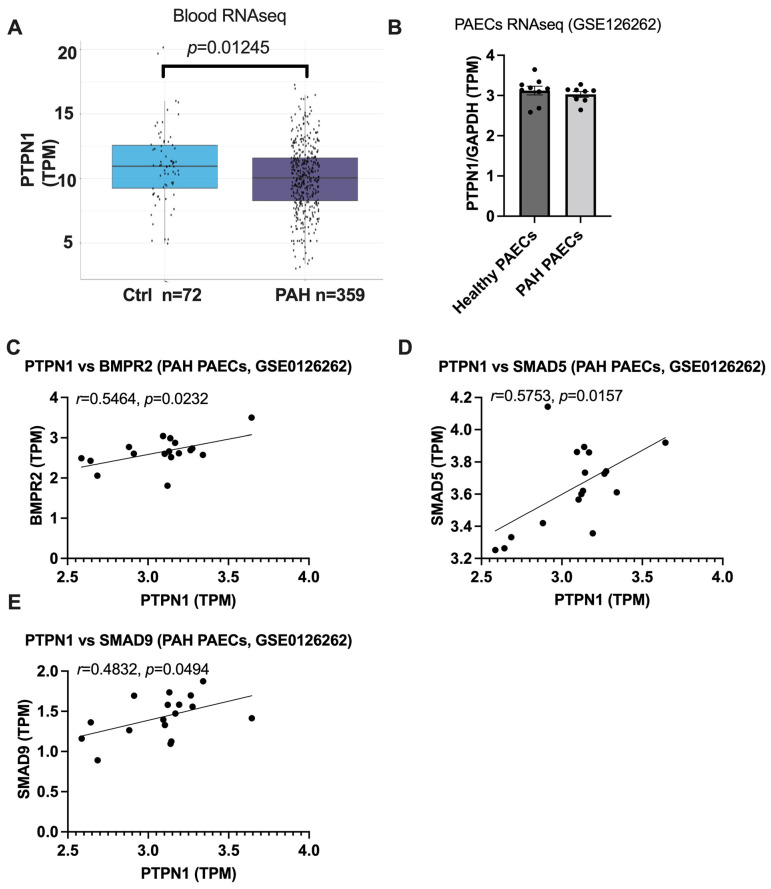
**PTPN1 is downregulated in the blood but not in the PAECs of PAH patients.** We analyzed RNA-seq data of PTPN1 expression in the whole blood ((n = 72 healthy and n = 359 PAH), for subject characteristics, please see [22]) (**A**) and in PAECs (n = 9/group, (GSE0126262, [23])) of PAH patients (**B**). TPM values for PTPN1 expression is shown in the graphs. PTPN1 was correlated with expression of BMPR2, SMAD5, and SMAD9 in healthy and PAH PAECs (GSE0126262) (**C**–**E**).

**Figure 5 cells-12-00316-f005:**
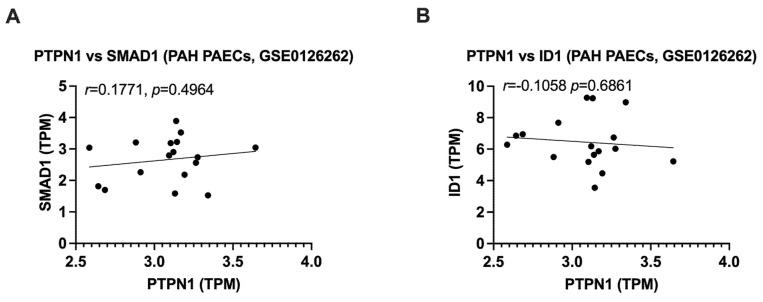
Correlation of PTPN1 expression with SMAD1 and ID1 in PAECs collected from healthy and PAH patients (GSE0126262). PTPN1 expression was not significantly correlated with SMAD1 (**A**) and ID1 (**B**) in the PAECs of PAH patients and healthy controls.

## Data Availability

Not applicable.

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
