# Peer review of "PTPN1 Deficiency Modulates BMPR2 Signaling and Induces Endothelial Dysfunction in Pulmonary Arterial Hypertension"

_cells, 2023, doi:10.3390/cells12020316_

Round 1

Reviewer 1 Report

The manuscript entitled "PTPN1 deficiency modulates BMPR2 signaling and induces endothelial dysfunction in Pulmonary Arterial Hypertension "identified PTPN1 as a novel regulator of BMPR2 signaling in pulmonary arterial endothelial cells . The finding improve our understanding of  molecular mechanisms of plmonary artery hypertension. The manuscript was also well written. 

Author Response

Reviewer 1:

Comment: The manuscript entitled "PTPN1 deficiency modulates BMPR2 signaling and induces endothelial dysfunction in Pulmonary Arterial Hypertension "identified PTPN1 as a novel regulator of BMPR2 signaling in pulmonary arterial endothelial cells. The finding improves our understanding of molecular mechanisms of pulmonary artery hypertension. The manuscript was also well written.

Response: We thank the reviewer for the positive appraisal of our manuscript.

Reviewer 2 Report

The manuscript by Ali et al identified a novel regulatory molecule PTPN1 for BMPR2 signaling by using siRNA-mediated high throughput screening. Silencing of PTPN1 in PAECs led to endothelial dysfunction in vitro. Relevance of PTPN1 in PAH patient was then confirmed by measuring its levels in blood and PAECs collected from PAH patients. Although some of the results are from reanalysis of previously published data, overall the discovery is novel and provides potential clinical relevance.

1.       The high throughput screening was done in myoblastoma cell line C2C12. Any poticular reason for not using endothelial cells?

2.       To be sure that the reduced ID1 transcription in PTPN1 silenced cells is due to reduced BMPR2 level, a rescue experiment by restoring BMPR2 to normal level is required.

3.       Please describe the figure panels rather than experimental procedures in the caption for figure2. No information can be found on from what experiments are the two graphs in 2C generated. Better use actual length (um) of the tube formation rather than pixels.

4.       Is there any effects on migration upon PTPN1 silencing in PAECs?

5.       It is actually difficult to know whether the effects observed in PTPN1 cells are due to general disturbance of signaling cascade or it is specific to BMPR2 signaling. Since the classic VEGF signaling is through a cascade of tyrosine phosphorylation, it could help to better understand the effect by looking at VEGFR2 signaling in PTPN1 silenced cells.

6.       Figure 3B and C, the trend for reduced PTPN1 level in PH rat model is clear. Could statistical significance be achieved by increasing n?

7.       Figure 4B-E, in the text it says n=9/group, however in the graphs it shows only 8 data points for the PAH patient group. It is interesting that PTPN1 expression shows correlation with the expression of BMPR2, SMAD5 and SMAD9. It would be necessary to introduce some background of the patient samples e.g. age, sex, stage of the disease. Were the cell from the patients carrying BMPR2 mutation or not? Are there changes in SMAD5 and SMAD9 expression or phosphorylation in PTPN1 silenced cells?

8. The figure on page 10, which is labeled figure 1, is not referred anywhere in the text. In general the data presentation of the manuscript needs to be improved.

Author Response

Reviewer 2:

General comment: The manuscript by Ali et al identified a novel regulatory molecule PTPN1 for BMPR2 signaling by using siRNA-mediated high throughput screening. Silencing of PTPN1 in PAECs led to endothelial dysfunction in vitro. Relevance of PTPN1 in PAH patient was then confirmed by measuring its levels in blood and PAECs collected from PAH patients. Although some of the results are from reanalysis of previously published data, overall, the discovery is novel and provides potential clinical relevance.

Response to general comment: We thank the reviewer for the positive appraisal of our manuscript’s study design, analysis, and findings.

Comment (1): The high throughput screening was done in myoblastoma cell line C2C12. Any particular reason for not using endothelial cells?

Response (1): We thank the reviewer for the comment. The high throughput screening (HTS) was performed in an engineered mouse myoblastoma reporter cell line, where the Bone Morphogenetic Protein Response Element (BRE) of the ID1 promoter was linked to luciferase. Upon activation of the ID1 promotor and addition of Luciferin, the activity could be determined in a luminometer.  The C2C12 cell line provides a useful screening platform because of its bipotentiality, robust culture capacity, and adaptability to scalable and automated quantitative assays. The experiments were performed on 384 well plates and the inter and intraplate variability was kept at a minimum. The human primary pulmonary arterial endothelial cells used in our study are not feasible for the large screening assays. As shown in our previous work using the same reporter cell line to screen FDA approved drugs, the hits from the screen are then validated in vascular cells of interest, such as endothelial cells or smooth muscle cells.

Comment (2): To be sure that the reduced ID1 transcription in PTPN1 silenced cells is due to reduced BMPR2 level, a rescue experiment by restoring BMPR2 to normal level is required.
Response (2): We thank the reviewer for this important comment and suggestion. Based on our findings, it seems ID1 downregulation (~50%) is not solely due to the downregulation of BMPR2 (15-20%) in PTPN1 silenced cells (Figures 1D-J). It could be due to the downregulation of SMAD signaling (SMAD1/5/9) or modulation of other factors, such microRNAs, transcription factors etc. As currently we do not know the exact molecular mechanisms of how PTPN1 regulates BMPR2 signaling, we acknowledged this limitation in the text, please see page 12, lines 347-350.

Given the likely multifactorial regulation of PTPN1, we do not believe that a rescue experiment would provide meaningful additional confirmation of the involvement of PTPN1 in BMPR2 signaling modification.

Comment (3): Please describe the figure panels rather than experimental procedures in the caption for figure2. No information can be found on from what experiments are the two graphs in 2C generated. Better use actual length (um) of the tube formation rather than pixels.
Response (3): We are thankful to the reviewer to point out these issues. We have now updated the figure legend with text modification.

We apologize for the confusion. The angiogenesis data were collected from the same experiment. The Y axis in the last panel of figure 2 would be “number of nodes in the analysed area” instead of “total tube length”. We have now described the figure panels appropriately and included the actual tube length in um instead of pixels in Page 16, Lines 488-493, and Figure 2C of the revised manuscript.

Comment (4): Is there any effects on migration upon PTPN1 silencing in PAECs?
Response (4): We did not perform migration assay following PTPN1 knockdown in PAECs.

Comment (5): It is actually difficult to know whether the effects observed in PTPN1 cells are due to general disturbance of signaling cascade or it is specific to BMPR2 signaling. Since the classic VEGF signaling is through a cascade of tyrosine phosphorylation, it could help to better understand the effect by looking at VEGFR2 signaling in PTPN1 silenced cells.

Response (5): We agree with the reviewer that the changes in endothelial cell function following PTPN1 knockdown could be due to either inhibiting BMPR2 signaling or modulating other signaling pathways. In fact, there is a strong crosstalk between BMPR2 signaling and the other signaling pathways, such as VEGF, ERK/MAPK, PI3K, AKT, p38, JNK, NOTCH, and TAK1. For details, please see our review manuscript Int J Mol Sci. 2018 Sep; 19(9): 2499.

We have acknowledged this in the discussion section lines 347-350, pages 12 of the revised manuscript.

Comment (6): Figure 3B and C, the trend for reduced PTPN1 level in PH rat model is clear. Could statistical significance be achieved by increasing n?

Response (6): We thank the reviewer for this comment and suggestion. Currently we do not have historical samples to increase n and repeat the experiment.

Comment (7): Figure 4B-E, in the text it says n=9/group, however in the graphs it shows only 8 data points for the PAH patient group. It is interesting that PTPN1 expression shows correlation with the expression of BMPR2, SMAD5 and SMAD9. It would be necessary to introduce some background of the patient samples e.g., age, sex, stage of the disease. Were the cell from the patients carrying BMPR2 mutation or not? Are there changes in SMAD5 and SMAD9 expression or phosphorylation in PTPN1 silenced cells?

Response (7): We excluded data from one PAH patient due to the low quality of the RNA sequencing data compared to others PAH 8 samples. Thus, we got only n=8 in the PAH patients group instead of n=9. Subject characteristics: PAH patients: 2HPAH (BMPR2 mutation), 5 IPAH, 1 APAH (drug and toxin associated) and 1 Pulmonary veno-capillary disease, average age 35.78 years, 4male/5 female, average 6MWD 301.43 m. Healthy controls: average age 43.22 years, sex 5 male/3 female. We have now included these texts in the method section lines 173-177, page 6 of the revised manuscript.

Regarding the confirmation of the relationship between PTPN1 and SMAD5/9 expression, we did not able to perform additional experiment to address this comment. Please note that, due to the limited availability of appropriate samples (particularly PTPN1 silenced PAECs), it was not feasible to address this comment with new analyses within the given 10 days revision time.

Comment (8): The figure on page 10, which is labeled figure 1, is not referred anywhere in the text. In general, the data presentation of the manuscript needs to be improved.

Response (8): We thank the reviewer for pointing this out. This should be Figure E1. We have now updated the information and referred in the main text and clarified the data in page 9 line 258-260 of the revised version of the manuscript.

Round 2

Reviewer 2 Report

Given the very limited time for revision, the authors have adequately addressed my questions and the manuscript has been improved.